# Preparation and Properties of Hydrophobic Polyurethane Based on Silane Modification

**DOI:** 10.3390/polym15071759

**Published:** 2023-03-31

**Authors:** Yuxian Ma, Minghui Zhang, Wenhao Du, Shixiong Sun, Benbo Zhao, Yuan Cheng

**Affiliations:** 1School of Chemistry and Chemical Engineering, North University of China, Taiyuan 030051, China; 2Dezhou Industrial Technology Research Institute, North University of China, Dezhou 253034, China

**Keywords:** silane coupling agent, waterborne polyurethane, acrylate, hydrophobic modification

## Abstract

Waterborne coatings have obtained more and more attention from researchers with increasing concerns in environmental protection, and have the advantages of being green, environmentally friendly and safe. However, the introduction of hydrophilic groups leads to lower hydrophobicity and it is difficult to meet the requirements of complex application environments. Herein, we proposed an optimization approach of waterborne polyurethane (WPU) with vinyl tris(β-methoxyethoxy) silane (A172), and it was found that the surface roughness, mechanical properties, thermal stability and water resistance of WPU will be increased to a certain extent with the addition of A172. Moreover, the hydrophobicity of the coating film is best when the silicon content is 10% of the acrylic monomer mass and the water contact angle reaches 100°, which could exceed two-thirds of the research results in the last decade. Therefore, our study can provide some theoretical basis for the research of hydrophobic polyurethane coatings.

## 1. Introduction

In recent years, considering environmental protection issues, waterborne coatings have been developed rapidly to reduce the emission of volatile organic compounds (VOCs). Among them, thanks to other resins and its ease of modification, waterborne polyurethane (WPU) could be conveniently modified by various methods to enhance its performance due to its excellent compatibility with other resins, which makes WPU widely used. However, the addition of hydrophilic chain extenders in the synthesis process resulted in the poor water resistance of WPU, which is the main obstacle in its application [1,2].

The most common modification methods to improve the properties of WPU include the introduction of acrylate monomers, silicones, organofluorine and nanomaterials [3,4,5,6]. Among them, the approach by introducing acrylates in WPU obtain the most attention, which could combine the advantages of acrylates and WPU to optimize the performance of coatings. However, due to the limitation of the flexible chain segment of acrylate, it still fails to meet the expected requirements (such as the water resistance always being unsatisfactory), so it is often combined with other modified methods to improve the comprehensive performance of the product [7]. Silicones are commonly used to improve this situation because of their outstanding properties such as high water resistance and heat resistance [8,9]. However, the strong hydrophobicity of silicone makes polyurethane difficult to emulsify in water and the phase separation degree will gradually increase due to the different solubility parameters of silicone and polyurethane, which may result in a decrease of emulsion stability. In order to overcome this contradiction, a variety of methods have been used to design the structure of silicone, such as the introducing of silane coupling agents, polyhedral low-polysilioxane (POSS), polar groups and polyether modification [10,11,12,13]. Among them, polyether modification and the introduction of polar group modification are both aimed at enhancing the water resistance of WPU by increasing the polarity of silicone, which often fail to achieve the expected effect, while modification with POSS always faces with issue of expensive raw materials and a complicated preparation process, the result of which is that it is difficult to use in most cases.

As a relatively common modification method, silane coupling agent contains both organic groups and inorganic silicon atoms, the latter of which have a smaller specific surface area and lower surface energy and can endow coatings with better hydrophobicity, high temperature resistance, etc. [14]. The main methods of modification with silane coupling agents include end modification, side-chain modification and synergistic modification [15]. For example, Gurunathan [16] and Gaddam et al. [17], respectively, modified WPU by end modification with 3-aminopropyltrimethoxysilane (APTMS) and 3-aminopropyltriethoxysilane (APTES), which contain only one amino group. During the reaction, the polyurethane prepolymer is directly sealed and the alkoxy group at the other end are hydrolyzed and condensed to form a cross-linked structure and improve the hydrophobicity of the material. However, considering that the use of end modification will limit the addition of silane, which will improve the performance of WPU to a limited extent [18], the side-chain modification method has been gradually considered to modify the WPU. Fu [19] and Lei et al. [20], respectively, modified WPU by side-chain modification with 3-mercaptopropyltrimethoxysilane and (3-(2-aminoethyl)aminopropyl)trimethoxysilane (AEAPTMS). Compared to the end modification, this method has a higher cross-link density and a larger relative molecular mass, which makes the WPU have better hydrophobicity and lower water absorption. In addition, in order to improve the comprehensive performance and versatility of the coatings, the synergistic modification has also been gradually developed [21]. For example, Zhao [22] and Yan et al. [23], respectively, modified WPU by synergistic modification with *N*-(2-aminoethyl)-3-aminopropyltriethoxysilane (AATS) and 3-glycidylethoxypropyltrimethoxysilane (GPTMS) together with APTMS. Zhang [24] used 3-(2-aminoethylamino)propyl dimethoxymethylsilane (KH-602) as a silane coupling agent to modify WPU with cyclophosphamide (PNMPD) containing double hydroxyl groups in synergistic side groups to obtain products with high fire resistance. From the above, we can see that the silane coupling agent not only has a wide range of applications, but also has a variety of pathways to improve the performance of WPU, which has a very far-reaching development prospect.

Among them, the stability and chemical reactivity of ethylene-based silane compounds make them suitable for use as silane coupling agents, and they are generally welcomed by producers and users because of the easy availability of synthetic raw materials, their simpler preparation methods and lower production costs. In the performance of modified polyurethane coatings, on the one hand, the addition of silane provides a hydrophobic surface with low surface energy. On the other hand, the hydrolysis of siloxane produces organosilanol, which will not exist stably, and it will further condense to generate silicon–oxygen–silicon bonds (Si-O-Si), leading to an increase in the chemical cross-linking point within the emulsion and an increase in cross-link density, which has an enhanced effect on the dense surface layer of the coating film and ultimately improves the thermal stability and mechanical properties of the coating film and the water resistance of the polymer. For example, Jing et al. [25] modified WPU with vinyltriethoxysilane (A151) by free radical polymerization reaction. It was found that the water contact angle of WPU was increased to 97° with the addition of A151, and the mechanical properties and solvent resistance of the coating film were also improved to some extent. Li et al. [26] prepared acrylate emulsions with self-crosslinking structures by introducing vinyltrimethoxysilane (A171) to address the problem of the poor water resistance of acrylate emulsions. It was found that the addition of A171 enhanced the water resistance and thermal stability of the resin to a certain extent. However, this modification also has certain defects, such as the hydrolytic condensation reaction of alkoxy groups being difficult to control [27], which makes it difficult for WPU to form and stabilize storage.

Compared to the alkoxy (methoxy, ethoxy, etc.) in general coupling agents, the 2-methoxyethoxy of vinyltris(β-methoxyethoxy)silane (A172) has a larger molecular weight. The larger alkoxy structure makes its hydrolysis slower and more stable, which can solve the problem of the difficult-to-control hydrolysis condensation reaction of silane groups to some extent. Moreover, as a more special organosilane coupling agent, it has better solubility compared with other types of silane coupling agents because of its own molecular structure containing an ether-type structure, which makes its surface coating modification of hydrophilic inorganic fractions more adequate and efficient.

To the best knowledge of authors, no researchers have reported the use of A172 to modify WPU. Therefore, a series of silane-modified waterborne polyurethane acrylates (SWPUA) with hydrophobic groups in the chain segments, which were used in hydrophobic wood coatings fields, were prepared by selecting vinyltris(β-methoxyethoxy) silane (A172) as the modified monomer and combining it with acrylates for cross-linking modification of WPU, and the effects of different additions of A172 on the polyurethane properties were investigated. As expected, the polymerization emulsions could exhibit an excellent overall performance and the best hydrophobicity when the silicon content was 10% of the acrylic monomer mass, which could exceed two-thirds of the research results in the last decade. Accordingly, we believe that our study can provide some theoretical basis for the research of hydrophobicity of wood coatings.

## 2. Materials and Methods

### 2.1. Materials

Toluene diisocyanate (TDI, Tech) was purchased from Gansu Yinguang Chemical Industry Base Co. (Baiyin, China) Polyether diol (DL1000, Tech) was provided by Shandong Blue Star Dongda Co. (Zibo, China) 2,2-Dihydroxymethylpropionic acid (DMPA, Tech) was provided by Shenzhen Golden Tenglong Industrial Co. (Shenzhen, China). 1,4-Butanediol (BDO, Tech) was produced by Jining Huakai Resin Co. (Jining, China). Dibutyltin dilaurate (DBTDL, CP) was purchased from Aladdin Reagent Co. (Shanghai, China). Triethylamine (TEA, AR) and *N*,*N*-dimethylformamide (DMF, AR) were provided by Tianjin Zhiyuan Chemical Reagent Co. (Tianjin, China) Methyl methacrylate (MMA, AR), hydroxyethyl acrylate (HEA, AR), and vinyltris(β-methoxyethoxy)silane (A172, AR) were produced by Shanghai Maclean Biochemical Technology Co. (Shanghai, China) Butyl acrylate (BA, AR) was purchased from Fuchen Chemical Reagent Co. (Tianjin, China) Potassium persulfate (KPS, AR) was provided by Tianjin Hengxing Chemical Reagent Manufacturing Co. (Tianjin, China) Trimethylolpropane triacrylate (TMPTA, AR) was produced by RYOJI Ryosei.

### 2.2. Synthesis

#### 2.2.1. Synthesis of Waterborne Polyurethane Emulsion (WPU)

Firstly, a certain amount of TDI (34.2 g), DL-1000 (70.1 g) and DBTDL were added into a four-necked flask equipped with a thermometer, mechanical stirrer and reflux condenser at 90 °C until the -NCO content reached the theoretical value. Then, DMPA (6.0 g), BDO (5.7 g), and HEA (4.1 g) were added sequentially, and the reaction time was determined by measuring the -NCO content during all periods. After cooling to 40 °C, a measured amount of TEA was added to neutralize the reaction for 50 min. Finally, the WPU emulsion was obtained after emulsifying for 1 h with deionized water under vigorous stirring. The reaction equation of WPU was shown in Figure 1.

#### 2.2.2. Synthesis of Silane-Modified Waterborne Polyurethane Acrylate Emulsion (SWPUA)

Firstly, A certain amount of MMA, BA, HEA, TMPTA, silane coupling agent A172, deionized water and WPU emulsion were added into a four-necked flask equipped with a thermometer, mechanical stirrer and reflux condenser. After 30 min, KPS (dissolved with deionized water) was added and stirred for 10 min, and then pre-emulsion was obtained by filtration. Then, a certain amount of deionized water was weighed into a four-necked flask, and after the temperature of the system was increased to 80 °C, the pre-emulsion was transferred to a constant pressure-dispensing funnel and the flow rate was controlled so that the pre-emulsion was added dropwise within 3 h. Subsequently, the system temperature was raised to 85 °C and kept warm for 1 h, and then the post-initiator was added dropwise to the four-neck flask within 20 min. Finally, it was kept warm for 1 h, and the SWPUA emulsion was obtained by cooling and filtering. The reaction equation of SWPUA is shown in Figure 2 and the ratio of raw materials is listed in Table 1.

#### 2.2.3. Preparation of Coating Film

A certain amount of emulsion was weighed and spread into the PTFE plates, then the films were dried at room temperature for 24 h. At last, the required films were obtained after placed in a vacuum drying oven at 30 °C for 24 h.

#### 2.2.4. Preparation of Paint Film

According to “GB/T1727.92 General Preparation Method of Paint Film” [28], it is prepared by brush coating method. First, the veneer was sanded smooth along the texture direction and the surface was cleaned of wood chips. Then, the emulsion was evenly applied to the veneer in the direction of the grain and dried naturally at room temperature for 24 h and then in a vacuum drying oven at 30 °C for 24 h. Ultimately, the paint film was obtained.

### 2.3. Characterization

Fourier transform infrared spectrometer (FTIR) was employed to characterize the chemical structure of WPU and SWPUA films at the attenuated total reflection (ATR) mode with a scan range of 4000~400 cm^−1^ and a resolution of 4 cm^−1^.

The storage stability of WPU and SWPUA emulsions was tested in accordance with GB/T6753.3-1986 for a period of six months.

The centrifugal stability of WPU and SWPUA emulsions was tested by ultracentrifuge, and the centrifuge speed was set at 3000 r/min and the centrifugation time was 15 min.

A benchtop SEM (TM4000) was used to observe the longitudinal section of the coating film at a magnification of 500× and the surface morphology of the coating film at a magnification of 1000×.

The WPU and SWPUA emulsions were diluted to be 0.1 wt% solution with distilled water, and then the particle size and potential were measured with a nanoparticle size and zeta potential tester at 25 °C. Each sample was scanned three times, and the results were averaged.

A mechanical testing machine was used to perform the tensile test of the specimens, which were cut into a dumbbell shape and subjected to stress–strain measurements at a strain rate of 5 mm/min. At least three measurements were performed for each specimen.

The thermal performance analysis of the coating films was measured with a thermogravimetric analyzer. The test conditions were from 25 °C to 600 °C at a heating rate of 10 °C/min under N_2_ atmosphere.

The contact angle of water on paint film was measured with a contact angle measuring instrument according to GB/T 30693-2014.

The prepared coating films were cut into pieces with size of 20 × 20 × 1 mm^3^ and immersed into deionized water at 25 °C for 24 h. The water absorption of the film was calculated by the following formula:(1)$$\mathrm{Water}\;\mathrm{absorption}=\frac{m_{1}-m_{0}}{m_{0}}\;\times \;100\%$$
where m_0_ is the mass of dried film and m_1_ is the mass of the film after being put into the water for 24 h.

## 3. Results and Discussion

### 3.1. FT-IR Analysis

The FTIR spectra of WPU and SWPUA are shown in Figure 3. As can be seen from Figure 3, according to the FTIR spectra before and after the modification, the characteristic peaks of NH stretching vibration and C=O stretching vibration, respectively, appeared at 3440 cm^−1^ and 1730 cm^−1^, and the absorption peak of 1240 cm^−1^ belonged to the asymmetric stretching vibration of C-O-C in the urethane group, and the stronger absorption peaks of -CH_3_ and -CH_2_- stretching vibration appeared at 2963 cm^−1^ and 2922 cm^−1^, which together indicated that the urethane group was formed in the system. In addition, the characteristic absorption peak of -NCO disappeared at 2270 cm^−1^, implying that the NCO groups have completely participated in the reaction.

Compared with the spectra of WPU, it can be found that the absorption peak of Si-C stretching vibration appeared at 995 cm^−1^ in the spectra of SWPUA, and the absorption peak at 1240 cm^−1^ was enhanced and broadened, indicating that in addition to the C-O-C stretching vibration, a Si-O stretching vibration also appeared, and the characteristic absorption peak of C=C disappeared at 3000–3100 cm^−1^, which indicated that free radical polymerization has occurred and the double bond had completely reacted, i.e., the silane coupling agent had been successfully introduced to the WPUA chain segment and the SWPUA emulsion had been prepared successfully [29]. In addition, it can be seen in the infrared spectrogram of SWPUA that stretching vibration absorption peak of Si-O-Si appeared at 1100 cm^−1^, which indicates the hydrolysis of alkoxy group in silane coupling agent A172. The organosilanol was produced because of the hydrolysis of the silane coupling agent, which will not exist stably, and it will further condense to generate siloxane bond (Si-O-Si).

### 3.2. Surface Morphology Analysis

The longitudinal images of WPU and SWPUA coating films at a SEM magnification of 500 and the surface images at a magnification of 1000 are shown in Figure 4.

As shown in Figure 4a_1_,a_2_, striated structures with more directional distribution appeared on the longitudinal section of WPU, while more scale-like structures appeared on the longitudinal section of SWPUA. Moreover, the scale-like structure of SWPUA was bigger and more disordered. The reason for this structural change may be that the addition of silicon destroys the regularity of the polyurethane structure and increases the degree of microphase separation of polyurethane [30].

Comparing the surface images of WPU Figure 4b_1_ and SWPUA Figure 4b_2_, it can be found that the surface of SWPUA coating film was rougher than WPU, which showed finer graininess on the image. This is because of the migration of silicone with low surface tension to the surface of the coating film after the addition of A172 [31]. Besides, the results were in good agreement with the hydrophobic results of the coating film.

### 3.3. Particle Size and Stability Analysis

Compared to the unmodified emulsion, the particle size of the emulsion increased with the addition of A172, which could be found from Figure 5. This is because the addition of A172 increases the molecular weight of the system, and the increase of molecular weight leads to an increase of the particle size of the system [32]. In addition, with the increase of the content of A172, the average particle size of the emulsion showed a trend of first decreasing and then increasing. This is because the introduction of unsaturated bonded siloxanes enhances the interaction between the polyurethane and acrylate chains, resulting in tighter intermolecular connections. As a result, the average particle size of the emulsion particles was reduced. However, when the grafting amount of the silane coupling agent reached more than 10% of the acrylate monomer mass, the particle size and PDI tended to increase. The reason for this phenomenon may be that an increase in vinyl siloxane content leads to an increase in the unsaturation of the system, which leads to cross-linking and agglomeration between emulsion particles, resulting in an increase in particle size and irregular distribution, ultimately leading to an increase in the average particle size and PDI of the emulsion with an increase in siloxane content [33].

It can be found from Figure 5 that the zeta potential showed a gradual decrease trend with the addition of certain silane coupling agents. This is because during the emulsion polymerization process, the hydrophobic silane coupling agent enters into the emulsion particles along with the acrylate monomer to form the inner core, which leads to the swelling of the emulsion particles and the reduction of the density of ionic groups on the surface of the polymer particles, eventually making the repulsive force between the polymer particles decrease. Moreover, the zeta potential is a measure of the strength of the interaction between the particles, so the zeta potential gradually decreases.

As shown in Appendix A (in Appendix A), when the addition of A172 was higher than 10% of the initial acrylate monomer mass, the storage stability of the emulsion decreased to less than 6 months, and precipitation occurred by centrifugation. On the one hand, this is because with the increase of the content of silane coupling agent, the cross-linking point and the cross-linking degree in the polymer molecular system increase, which leads to the gradual change of molecular chain from linear to bulk, and the intermolecular movement changes from intermolecular forces to chemical bonds, so the resistance of movement becomes larger. Therefore, the stability decreased and precipitation occurred. On the other hand, during the long-term storage process, the emulsion particles settle under the action of gravity and form a concentrated layer at the bottom of the container, which makes a reduction in the spacing of the emulsion particles, and some of the particles cross the potential barrier and become unstable and coalesce. In addition, the larger the particle size of the emulsion, the faster the settling velocity of the emulsion particles, which is detrimental to the storage stability of the emulsion.

### 3.4. Mechanical Performance Analysis

The stress–strain curves of polyurethane coating films with different silane coupling agent additions are shown in Figure 6a, and their maximum tensile strength and elongation at break with silicon content are shown in Figure 6b. The specific data is shown in Table 2.

According to Figure 6 and Table 2, The mechanical properties of WPU were improved to some extent with the addition of A172, and the tensile strength of the films gradually increased with the increase of the content of silane coupling agent A172, and the elongation at break showed a trend of increasing and then decreasing [25,26,27,28]. The reason why the tensile strength of SWPUA increases is probably because when A172 comes into WPU molecule, it can effectively increase the cross-link density of the system, which makes the intermolecular force enhanced and restricts the movement of the molecular chain. When small contents of A172 were added, the elongation at break increased. This is probably because the addition of a small amount of A172 increases the degree of microphase phase separation between hard and soft segments, which makes the intermolecular arrangement less tight. Therefore, the elongation at break increases. However, when the contents of A172 continues to increase, the elongation at the break decreased because when the silane coupling agent was added to a certain concentration, the cross-linking effect plays a major role compared to the microphase separation effect, which would restrict the movement of molecular chains; thus, the elongation at the break tended to decrease [34,35,36,37].

### 3.5. TG Analysis

The TG curves and DTG curves of waterborne polyurethane with different silicon content are shown in Figure 7a,b, respectively. As seen in Figure 7, the thermal decomposition process of waterborne polyurethane film was mainly divided into three stages. In general, the first degradation stage in the temperature ranged from 0 to 200 °C, which was mainly caused by the decomposition of small molecules and the evaporation of water. The second stage of decomposition ranged from 250 to 350 °C, which was mainly the decomposition of the carbamate and urea bonds in the hard chain segment. Finally, the last stage ranged from 350 to 450 °C, which was the decomposition of ether bond and silica bond in the soft chain sections.

As shown in the figure, the thermal stability of WPU was improved to some extent with the addition of A172. With the introduction of silane coupling agent, the temperature at 50% mass loss (T50%) and final residue mass of the modified specimens all increased. Moreover, the final residue mass of coating films increased with the increase of silicon content. On the one hand, because of the introduction of silicon–oxygen bond, the bond energy of Si-O bond is higher than C-C and C-O bonds [38,39], which makes it more difficult to break (Si-O bond energy is 550 KJ/mol, C-C or C-O bond energy is 340 KJ/mol). On the other hand, because of the double bond of A172, the increase in the content of the double bond leads to the increase in the degree of cross-linking, and the chemical bonds between polyurethane and polyacrylate molecules also enhance the intermolecular interactions. The synergistic effect of the two ultimately leads to the increasing of thermal stability of the coating film. Therefore, the addition of silicon can effectively increase the thermal stability of the WPU coating film.

### 3.6. Water Absorption and Water Contact Angle Analyses

As seen in Figure 8a, with the addition of silane coupling agent, the water resistance of WPU coating film has been improved to a certain extent, which is intuitively shown by the decrease of the water absorption rate. With the increase of the silicon content, the water absorption of the coating film showed a trend of gradual decrease. The reason for this is that the addition of silane coupling agent leads the increase of the cross-link density of the system, which makes it difficult for hydrophilic groups to move to the surface of the film and further prevents water molecules from penetrating into the interior of the film; thus, the water absorption of the coating film decreases.

As shown in Figure 8b, with the addition of A172, the hydrophobicity of WPU was improved to a certain extent, which was reflected by the increase of water contact angle. With an increase in the content of A172 in SWPUA film from 0 to 20%, the water contact angle of SWPUA coating film first increased obviously and then decreased. Among them, when the A172 content was added at 10% of the acrylic monomer mass, the water contact angle reached a maximum value of 100°. The reason why the water contact angle tends to increase first is because of the migration of silicon to the surface of the coating film and the outward orientation of the side-chain silanes of the polymer, which reduces the surface energy of the coating film and corresponds to the formation of a hydrophobic surface with low surface energy; thus, the water contact angle and hydrophobicity increase. However, when the silicon content exceeds 10% of the acrylic monomer mass, the water contact angle decreases, because as the silicon content continues to increase, on the one hand, the effective enrichment of silicon on the surface reaches the maximum value, a critical micelle concentration appears and the reduction of surface energy gradually tends to level off, so the water contact angle no longer increases. On the other hand, when the degree of cross-linking of the polymer increases to a certain extent, the potential resistance effect is obvious, which will further reduce the polymeric grafting of polyurethane and acrylate resin; thus, the water contact angle decreases [40,41]. Therefore, the addition of a certain amount of A172 can enhance the surface hydrophobicity of coatings to a certain extent. In the last decade, there are few studies on enhancing the hydrophobicity of polyurethane with silicon modification, and some data on hydrophobicity studies of polyurethane in the last decade at home and abroad were briefly listed in Figure 9 [12,18,20,25,30,31,42,43,44,45,46,47,48,49,50,51,52,53,54,55].

It can be found from Figure 9 that the optimal addition amounts of silicon for each system varied from one research work to another because of the different original formulation and the type of silicon modification. It can be found that the water contact angle of the modified coating film is mostly concentrated between 80 and 100° so far. It is generally accepted that when the water contact angle is greater than 90°, the coating film is considered to have good hydrophobicity. Therefore, in comparison with other research works, it is clear that our findings are at an advanced level, exceeding two-thirds of the research results.

## 4. Conclusions

In this study, a series of modified waterborne polyurethane acrylates (SWPUA) with different silicon contents were prepared by cross-linking the WPU with vinyl tris(β-methoxyethoxy)silane (A172), which will be applied to the use of hydrophobic wood coatings. The results showed that the addition of A172 led to different degrees of improvement of hydrophobicity, water resistance, mechanical properties and thermal stability of WPU, among which the improvement of hydrophobicity was the most significant. With the addition of silane coupling agent, the particle size of emulsion showed a trend of increasing, then decreasing and then increasing again, and the Zeta potential showed a decreasing trend. It was found that the emulsion modified with A172 could be stably stored for 6 months. Scanning electron microscopy showed that the addition of silane coupling agent increased the surface roughness and the degree of microphase separation of the coating film. After the coating film formation, the tensile test and TG test data showed that the mechanical properties and thermal stability of the coating were improved. The water absorption test revealed that the water absorption of the coating film gradually decreased. The hydrophobicity test of SWPUA showed that the water contact angle of the film tended to increase first and then slightly decrease. Under the condition of stable storage of the emulsion, when the addition of A172 was 10% of the acrylate mass, the water contact angle of SWPUA film was increased from 71° to 100°, which could exceed two-thirds of the research results at home and abroad in the last decade. The results showed that without affecting the storage stability of the emulsion, the addition of A172 could solve the problem of poor hydrophobicity of waterborne polyurethane to some extent. In addition, it can improve the comprehensive performance of polyurethane to a certain extent.

## Figures and Tables

**Figure 1 polymers-15-01759-f001:**
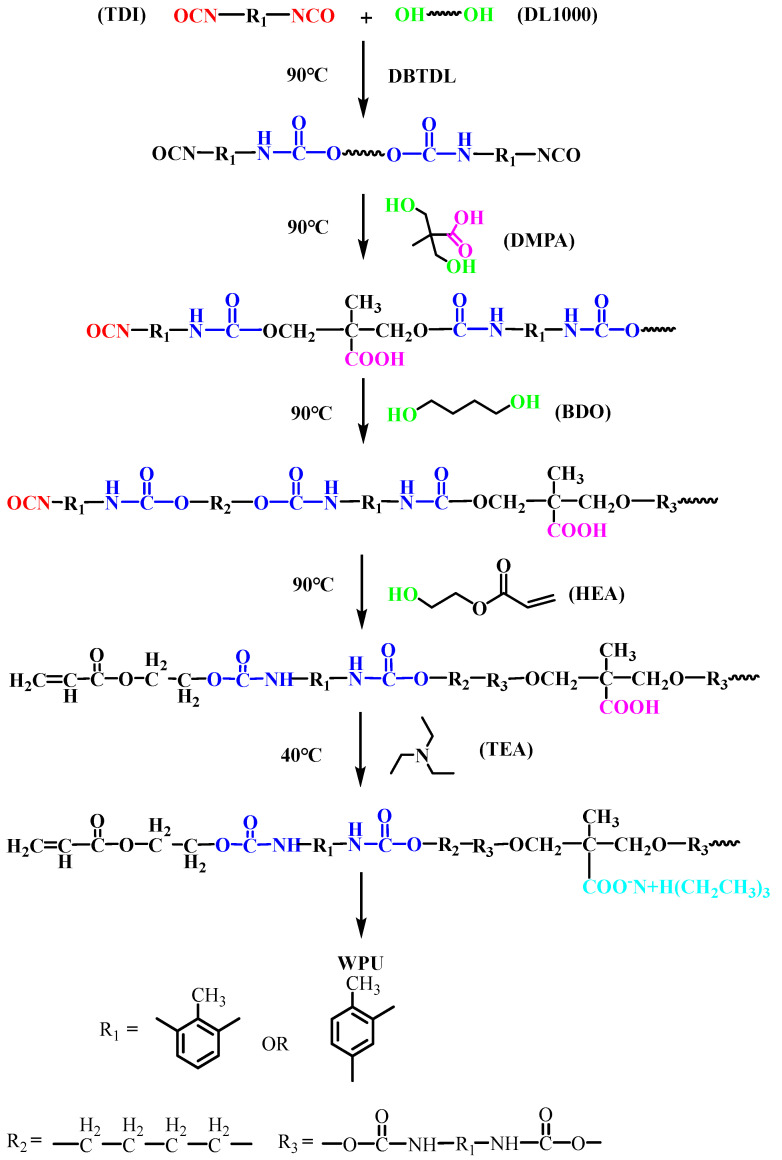
Synthetic route of WPU. The groups with colors in the figure represent the groups involved in the reaction. Red represents −NCO, green represents −OH, blue represents −NHCOO, purple represents −COOH, and cyan represents −COO^−^N^+^H(CH_2_CH_3_)_3_.

**Figure 2 polymers-15-01759-f002:**
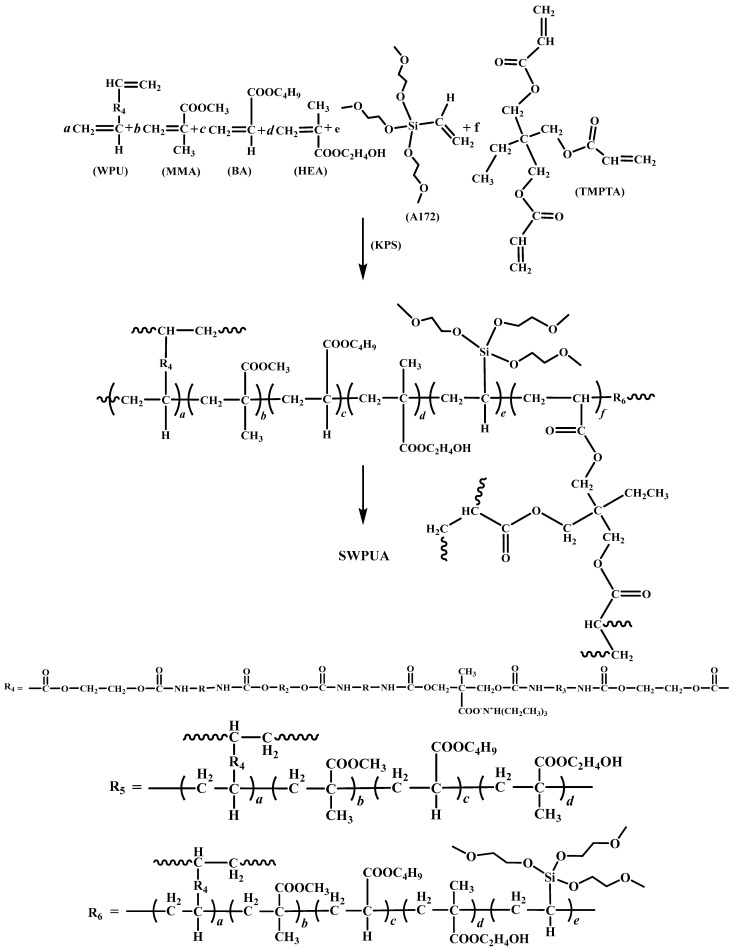
Synthetic route of SWPUA.

**Figure 3 polymers-15-01759-f003:**
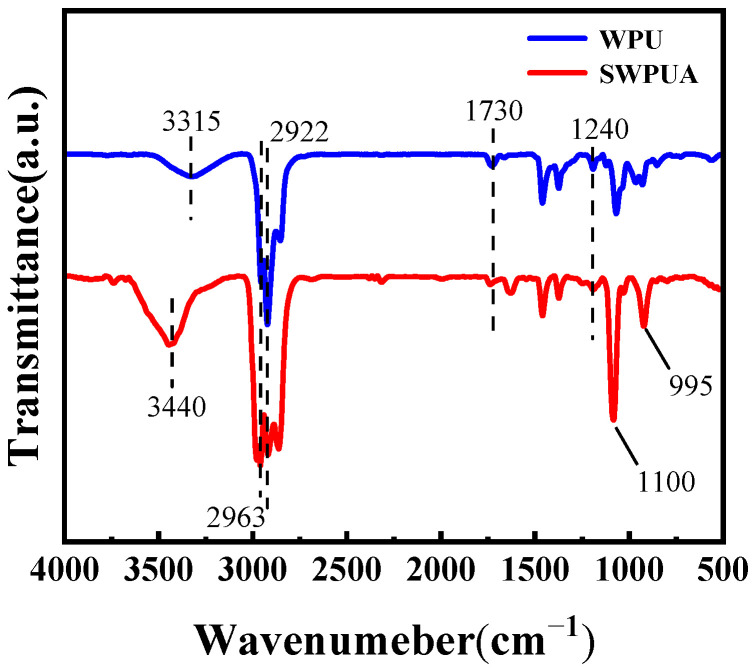
FT-IR spectra of WPU and SWPUA.

**Figure 4 polymers-15-01759-f004:**
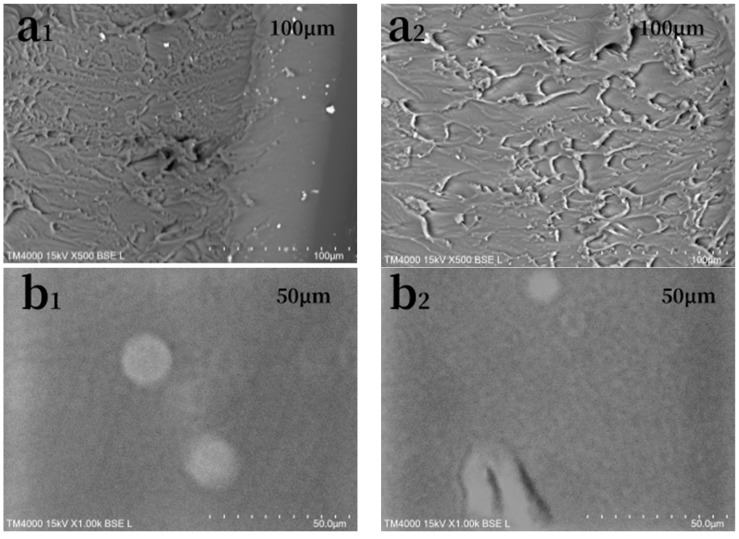
TM4000 images of WPU longitudinal section (**a_1_**), WPU surface (**b_1_**), SWPUA longitudinal section (**a_2_**), SWPUA surface (**b_2_**).

**Figure 5 polymers-15-01759-f005:**
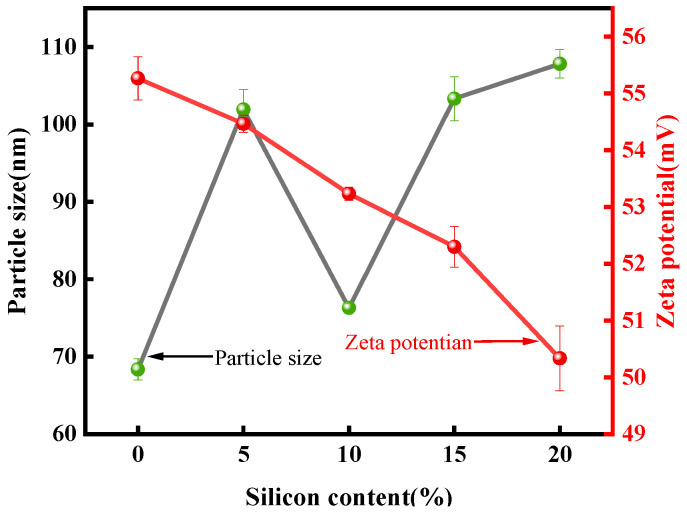
Particle size and zeta potential of SWPUA emulsion with different silicon content.

**Figure 6 polymers-15-01759-f006:**
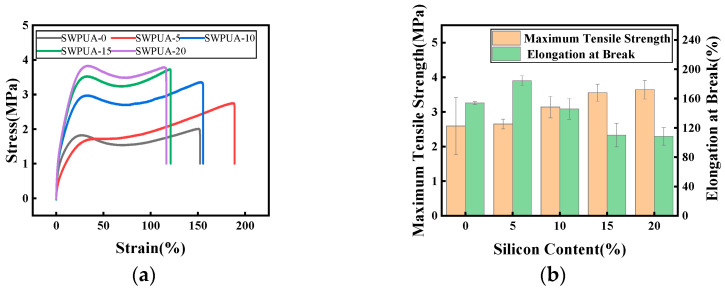
(**a**) Stress–Strain curves of SWPUA samples with different silicon content; (**b**) Maximum tensile strength and Elongation at break of SWPUA samples with different silicon content.

**Figure 7 polymers-15-01759-f007:**
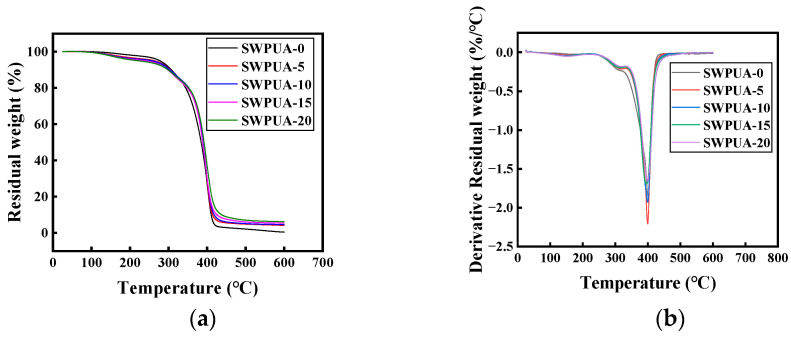
(**a**) Weight loss curves of SWPUA samples with different silicon content; (**b**) Derivative of weight loss curves of SWPUA samples with different silicon content.

**Figure 8 polymers-15-01759-f008:**
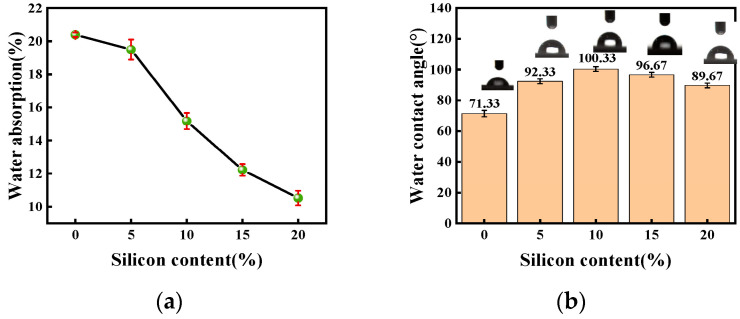
(**a**) Water absorption of SWPUA samples with different silicon content; (**b**) Water contact angle of SWPUA samples with different silicon content.

**Figure 9 polymers-15-01759-f009:**
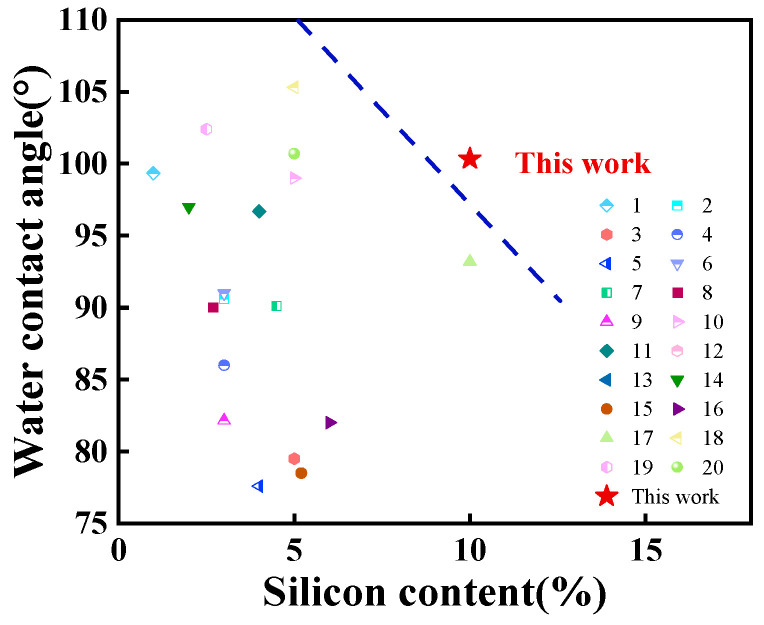
The maximum water contact angle of different research systems.

**Table 1 polymers-15-01759-t001:** The experimental formulation of SWPUA.

WPU	MMA	BA	HEA	TMPTA	A172	
g	g	g	g	g	ω_A172_/%	g
100	39.9	18.6	1.5	1.2	0	0
100	39.9	18.6	1.5	1.2	5	3
100	39.9	18.6	1.5	1.2	10	6
100	39.9	18.6	1.5	1.2	15	9
100	39.9	18.6	1.5	1.2	20	12

**Table 2 polymers-15-01759-t002:** The maximum tensile strength and elongation at break of SWPUA coatings with different silicon content.

Sample	SWPUA-0	SWPUA-5	SWPUA-10	SWPUA-15	SWPUA-20
Silicon content/%	/	5	10	15	20
Maximum tensile strength/MPa	2.59 ± 0.82	2.65 ± 0.14	3.14 ± 0.31	3.56 ± 0.25	3.64 ± 0.27
Elongation at break/%	154 ± 2	184 ± 7	146 ± 14	110 ± 16	108 ± 12

## Data Availability

The data presented in this study are available on request from the corresponding author.

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
