# Peer review of "Preparation and Properties of Hydrophobic Polyurethane Based on Silane Modification"

_polymers, 2023, doi:10.3390/polym15071759_

Round 1

Reviewer 1 Report

The manuscript is entitled " Preparation and properties of hydrophobic polyurethane based on silane modification". Authors have reported the preparation of modified waterborne polyurethane to provide a basis for the research of hydrophobic polyurethane coatings. However, there are some points that need to be corrected. Therefore, recommended the publication of this paper after minor revision.

Abstract

Paragraphs should not begin with connectors such as with. Instead, recheck the sentences for an effective and satisfactory abstract.

The study’s analyses should be briefly stated in the abstract section. Please make the necessary changes.

Introduction

The purpose of the study should be stated to the reader in a separate paragraph at the end of the introduction. Please make the necessary changes.

Materials and Methods

Line 111,113,141: Please specify the open states of DL-1000, DMPA, and PTFE.

Schemes should be changed to figures.

It may be helpful for the reader to provide a reference to the "General preparation method of the paint film."

Line 143,149: Please correct the spelling of degrees.

Result

Should the figure number on line 184 be 3? Please check again.

Check, in line 225; The is because….

You can specify what the red and green colors in Figure 3 represent in the figure. This will help make it more understandable in terms of meaning.

What do the numbers 1-23 in Figure 7 represent? These should be expressed clearly.

Reviewer 2 Report

The paper "Preparation and properties of hydrophobic polyurethane based on silane modification" reports on the synthesis and evaluation of structural, morphological, thermal, wettability and mechanical properties of new waterborne polyurethanes possessing silane groups.

The composition of the manuscript is well presented and the methods used for the preparation of polyurethanes are well performed.

I have the following suggestions:

-The Abstract must be rewritten; there is a lack of information related to the content of the paper.

-The Introduction must highlight better the advantages of the silane-based additives, and thus to emphasize better the novelty and the aim of this paper. Why vynil-containing silanes are more suitable for coating applications than other systems? Some reports regarding this field must be added.

-In Section 2.2 a table containing the amounts of each component, as well the molar ratio between monomers must be added.The preparation of SWPUA is not clear presented, the amounts are also necessary to be mentioned. Usually, the alkoxy groups at the silicon atom are hydrolyzable leading to siloxane bonds formation. As presented in Scheme 2, the final structure of the product must be confirmed.

- critical micellar concentration must be added for both systems in order to understand the aggregation behavior.

-in Section 3.1. - the main concern is related to the the presence of the silane units. If a crosslinked network is formed, most probably siloxane bonds are generated by hydrolysis of alkoxy groups, so some improvements are needed in IR disscusion.

-Section 3.3- Fig. 3 must be deleted, the information is presented in Table 2. -The analysis of WPU must be added to compare the influence of the presence of silane units in sections 3.4, 3.5 and 3.6.

-in Conclusions the main comparisons with WPU must be added.

Based on these observations my recommendation is Major Revision.

Reviewer 3 Report

The article analysis waterborne polyurethane coatings with silicone. Hydrophobicity, thermal stability, mechanical performance and structural changes were analysed. My remarks for this article are presented below:

1. The Abstract part should be improved with numerical values of the main findings of the study because this is the part which is firstly read to decide whether the article is worth reading.

2. I would like to suggest increasing the scaling of Scheme 1 and Scheme 2 as the writings on it can be barely seen.

3. Figure 2. The scale bar cannot be seen. I would suggest putting in on the image.

4. Figure 4,5,6. Suggest increasing the scaling of at least x and y axes and titles.

5. Even though the 3rd section is called Results and Discussion, I cannot find any discussion with other authors works in the same field. I would suggest improving this part.

6. Article should be formatted according to journal's template, i.e. the main text. 

7. Also, I would suggest thinking of compositions abbreviations in all graphs and Tables because A, B, C, D, etc. do not say anything about the composition of polyurethane or content of silicone. It is quite frustrating to go back to the beginning of the article to find what those letters mean.

8. Elongation at break is expressed in %, so the question is it really necessary to present such accuracy of the results? Also, where are upper and lower limits or standard deviations of the results?

9. Figure 6b. Why contact angle is presented in such accuracy? Why do you need this? What such accuracy shows?

10. Can you explain please why water absorption decreases with the addition of silicone while contact angle increases only up to 10 wt% silicone? Shouldn't it be the same tendency for both properties?

11. I would like to suggest deleting -5 in x axis of Figure 7. I do not think that the contact angle can be -5 deg. 

12. reference list should be modified according to journal's requirements.

Round 2

Reviewer 2 Report

The authors responded to all the suggestions and improved their work highlighting the novelty better, so that I recommend the publication of this paper in the present form.

Reviewer 3 Report

Authors have taken into consideration all my remarks and answered all my questions.